# DYNAMIC SPARSE GRAPH FOR EFFICIENT DEEP LEARNING

**Liu Liu**[12*]**, Lei Deng**[2*]**, Xing Hu**[2]**, Maohua Zhu**[2]**, Guoqi Li**[3]**, Yufei Ding**[2]**, Yuan Xie**[1]

[1]Department of Electrical and Computer Engineering, University of California, Santa Barbara
[2]Department of Computer Science, University of California, Santa Barbara
[3]Center for Brain Inspired Computing Research,
  Department of Precision Instrument, Tsinghua University
[*]Equal contribution
{liu_liu, leideng, huxing, maohua, yuanxie}@ece.ucsb.edu
yufeiding@cs.ucsb.edu
liguoqi@mail.tsinghua.edu.cn

## ABSTRACT

We propose to execute deep neural networks (DNNs) with dynamic and sparse graph (DSG) structure for compressive memory and accelerative execution during both training and inference. The great success of DNNs motivates the pursuing of lightweight models for the deployment onto embedded devices. However, most of the previous studies optimize for inference while neglect training or even complicate it. Training is far more intractable, since (i) the neurons dominate the memory cost rather than the weights in inference; (ii) the dynamic activation makes previous sparse acceleration via one-off optimization on fixed weight invalid; (iii) batch normalization (BN) is critical for maintaining accuracy while its activation reorganization damages the sparsity. To address these issues, DSG activates only a small amount of neurons with high selectivity at each iteration via a dimension-reduction search and obtains the BN compatibility via a double-mask selection. Experiments show significant memory saving (1.7-4.5x) and operation reduction (2.3-4.4x) with little accuracy loss on various benchmarks.

## 1 INTRODUCTION

Deep Neural Networks (DNNs) (LeCun et al., 2015) have been achieving impressive progress in a wide spectrum of domains (Simonyan & Zisserman, 2014; He et al., 2016; Abdel-Hamid et al., 2014; Redmon & Farhadi, 2016; Wu et al., 2016), while the models are extremely memory- and compute-intensive. The high representational and computational costs motivate many researchers to investigate approaches on improving the execution performance, including matrix or tensor decomposition (Xue et al., 2014; Novikov et al., 2015; Garipov et al., 2016; Yang et al., 2017; Alvarez & Salzmann, 2017), data quantization (Courbariaux et al., 2016; Zhou et al., 2016; Deng et al., 2018; Leng et al., 2017; Wen et al., 2017; Wu et al., 2018; McKinstry et al., 2018), and network pruning (Ardakani et al., 2016; Han et al., 2015b;a; Liu et al., 2017; Li et al., 2016; He et al., 2017; Luo et al., 2017; Wen et al., 2016; Molchanov et al., 2016; Sun et al., 2017; Spring & Shrivastava, 2017; Lin et al., 2017a; Zhang et al., 2018; He et al., 2018a; Chin et al., 2018; Ye et al., 2018; Luo & Wu, 2018; Hu et al., 2018; He et al., 2018b). However, most of the previous work aim at inference while the challenges for reducing the representational and computational costs of training are not well-studied. Although some works demonstrate acceleration in the distributed training (Lin et al., 2017b; Goyal et al., 2017; You et al., 2017), we target at the single-node optimization, and our method can also boost training in a distributed fashion.

DNN training, which demands much more hardware resources in terms of both memory capacity and computation volume, is far more challenging than inference. Firstly, activation data in training will be stored for backpropagation, significantly increasing the memory consumption. Secondly, training iteratively updates model parameters using mini-batched stochastic gradient descent. We almost always expect larger mini-batches for higher throughput (Figure 1(a)), faster convergence, and better accuracy (Smith et al., 2017). However, memory capacity is often the limitation factor (Figure 1(b))

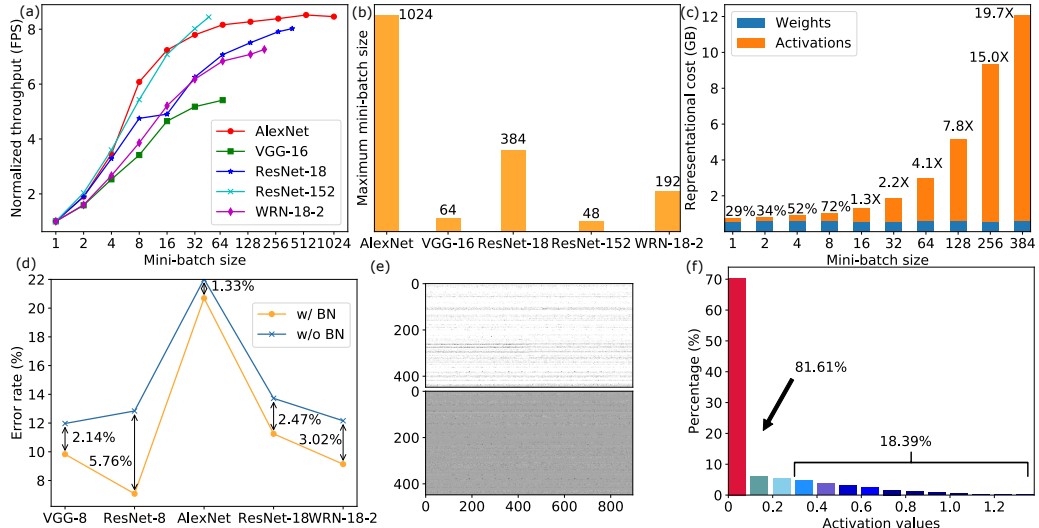

Figure 1: Comprehensive motivation illustration. (a) Using larger mini-batch size helps improve throughput until it is compute-bound; (b) Limited memory capacity on a single computing node prohibits the use of large mini-batch size; (c) Neuronal activation dominates the representational cost when mini-batch size becomes large; (d) BN is indispensable for maintaining accuracy; (e) Upper and lower one are the feature maps before and after BN, respectively. However, using BN damages the sparsity through information fusion; (f) There exists such great representational redundancy that more than 80% of activations are close to zero.

that may cause performance degradation or even make large models with deep structures or targeting high-resolution vision tasks hard to train (He et al., 2016; Wu & He, 2018).

It is difficult to apply existing sparsity techniques towards inference phase to training phase because of the following reasons: 1) Prior arts mainly compress the pre-trained and fixed weight parameters to reduce the off-chip memory access in inference (Han et al., 2016; 2017), while instead, the dynamic neuronal activations turn out to be the crucial bottleneck in training (Jain et al., 2018), making the prior inference-oriented methods inefficient. Besides, during training we need to stash a vast batched activation space for the backward gradient calculation. Therefore, neuron activations creates a new memory bottleneck (Figure 1(c)). In this paper, we will sparsify the neuron activations for training compression. 2) The existing inference accelerations usually add extra optimization problems onto the critical path (Wen et al., 2016; Molchanov et al., 2016; Liu et al., 2017; Luo et al., 2017; Liang et al., 2018; Zhang et al., 2018; Hu et al., 2018; Luo & Wu, 2018; Ye et al., 2018), i.e., 'complicated training ⇒ simplified inference', which embarrassingly complicates the training phase. 3) Moreover, previous studies reveal that batch normalization (BN) is crucial for improving accuracy and robustness (Figure 1(d)) through activation fusion across different samples within one mini-batch for better representation (Morcos et al., 2018; Ioffe & Szegedy, 2015). BN almost becomes a standard training configuration; however, inference-oriented methods seldom discuss BN and treat BN parameters as scaling and shift factors in the forward pass. We further find that BN will damage the sparsity due to the activation reorganization (Figure 1(e)). Since this work targets both training and inference, the BN compatibility problem should be addressed.

From the view of information representation, the activation of each neuron reflects its selectivity to the current stimulus sample (Morcos et al., 2018), and this selectivity dataflow propagates layer by layer forming different representation levels. Fortunately, there is much representational redundancy, for example, lots of neuron activations for each stimulus sample are so small and can be removed (Figure 1(f)). Motivated by above comprehensive analysis regarding memory and compute, we propose to search critical neurons for constructing a sparse graph at every iteration. By activating only a small amount of neurons with a high selectivity, we can significantly save memory and simplify computation with tolerable accuracy degradation. Because the neuron response dynamically changes under different stimulus samples, the sparse graph is variable. The neuron-aware dynamic and sparse graph (DSG) is fundamentally distinct from the static one in previous work on permanent weight pruning since we never prune the graph but activate part of them each

time. Therefore, we maintain the model expressive power as much as possible. A graph selection method, dimension-reduction search, is designed for both compressible activations with element-wise unstructured sparsity and accelerative vector-matrix multiplication (VMM) with vector-wise structured sparsity. Through double-mask selection design, it is also compatible with BN. We can use the same selection pattern and extend our method to inference. In a nutshell, we propose a compressible and accelerative DSG approach supported by dimension-reduction search and double-mask selection. It can achieve 1.7-4.5x memory compression and 2.3-4.4x computation reduction with minimal accuracy loss. This work simultaneously pioneers the approach towards efficient online training and offline inference, which can benefit the deep learning in both the cloud and the edge.

## 2 APPROACH

Our method forms DSGs for different inputs, which are accelerative and compressive, as shown in Figure2(a). On the one hand, choosing a small number of critical neurons to participate in computation, DSG can reduce the computational cost by eliminating calculations of non-critical neurons. On the other hand, it can further reduce the representational cost via compression on sparsified activations. Different from previous methods using permanent pruning, our approach does not prune any neuron and the associated weights; instead, it activates a sparse graph according to the input sample at each iteration. Therefore, DSG does not compromise the expressive power of the model.

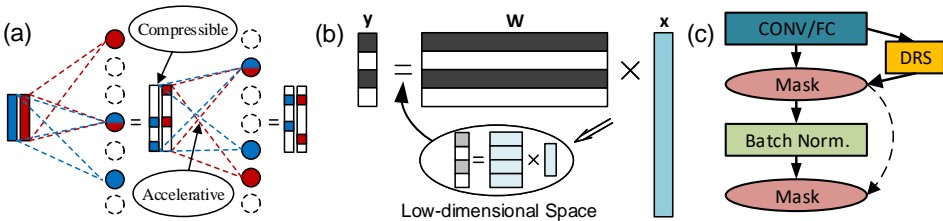

Figure 2: (a) Illustration of dynamic and sparse graph (DSG); (b) Dimension-reduction search for construction of DSG; (c) Double-mask selection for BN compatibility. 'DRS' denotes dimension-reduction search.

Constructing DSG needs to determine which neurons are critical. A naive approach is to select critical neurons according to the output activations. If the output neurons have a small or negative activation value, i.e., not selective to current input sample, they can be removed for saving representational cost. Because these activations will be small or absolute zero after the following ReLU non-linear function (i.e., ReLU$(x) = $ max$(0, x)$), it's reasonable to set all of them to be zero. However, this naive approach requires computations of all VMM operations within each layer before the selection of critical neurons, which is very costly.

### 2.1 DIMENSION-REDUCTION SEARCH

To avoid the costly VMM operations in the mentioned naive selection, we propose an efficient method, i.e., dimension reduction search, to estimate the importance of output neurons. As shown in Figure2(b), we first reduce the dimensions of $\mathbf{X}$ and $\mathbf{W}$, and then execute the lightweight VMM operations in a low-dimensional space with minimal cost. After that, we estimate the neuron importance according to the virtual output activations. Then, a binary selection mask can be produced in which the zeros represent the non-critical neurons with small activations that are removable. We use a top-$k$ search method that only keeps largest $k$ neurons, where an inter-sample threshold sharing mechanism is leveraged to greatly reduce the search cost [1]. Note that $k$ is determined by the output size and a pre-configured sparsity parameter $\gamma$. Then we can just compute the accurate activations of the critical neurons in the original high-dimensional space and avoid the calculation of the non-critical neurons. Thus, besides the compressive sparse activations, the dimension-reduction search can further save a significant amount of expensive operations in the high-dimensional space.

---

[1]Implementation details are shown in Appendix B.

Figure 3: Compressive and accelerative DSG. (a) Original dense convolution; (b) Converted accelerative VMM operation; (c) Zero-value compression.

In this way, a vector-wise structured sparsity can be achieved, as shown in Figure 3(b). The ones in the selection mask (marked as colored blocks) denote the critical neurons, and the non-critical ones can bypass the memory access and computation of their corresponding columns in the weight matrix. Furthermore, the generated sparse activations can be compressed via the zero-value compression (Zhang et al., 2000; Vijaykumar et al., 2015; Rhu et al., 2018) (Figure 3(c)). Consequently, it is critical to reduce the vector dimension but keep the activations calculated in the low-dimensional space as accurate as possible, compared to the ones in the original high-dimensional space.

## 2.2 SPARSE RANDOM PROJECTION FOR EFFICIENT DIMENSION-REDUCTION SEARCH

*Notations*: Each CONV layer has a four dimensional weight tensor $(n_K, n_C, n_R, n_S)$, where $n_K$ is the number of filters, i.e., the number of output feature maps (FMs); $n_C$ is the number of input FMs; $(n_R, n_S)$ represents the kernel size. Thus, the CONV layer in Figure 3(a) can be converted to many VMM operations, as shown in Figure 3(b). Each row in the matrix of input FMs is the activations from a sliding window across all input FMs ($n_{CRS} = n_C \times n_R \times n_S$), and after the VMM operation with the weight matrix ($n_{CRS} \times n_K$) it can generate $n_K$ points at the same location across all output FMs. Further considering the $n_{PQ} = n_P \times n_Q$ size of each output FM and the mini-batch size of $m$, the whole $n_{PQ} \times m$ rows of VMM operations has a computational complexity of $O(m \times n_{PQ} \times n_{CRS} \times n_K)$. For the FC layer with $n_C$ input neurons and $n_K$ output neurons, this complexity is $O(m \times n_C \times n_K)$. Note that here we switch the order of BN and ReLU layer from 'CONV/FC-BN-ReLU' to 'CONV/FC-ReLU-BN', because it's hard to determine the activation value of the non-critical neurons if the following layer is BN (this value is zero for ReLU). As shown in previous work, this reorganization could bring better accuracy (Mishkin & Matas, 2015).

For the sake of simplicity, we just consider the operation for each sliding window in the CONV layer or the whole FC layer under one single input sample as a basic optimization problem. The generation of each output activation $y_j$ requires an inner product operation, as follows:

$$y_j = \varphi(\langle \mathbf{X}_i, \mathbf{W}_j \rangle) \tag{1}$$

where $\mathbf{X}_i$ is the $i$-th row in the matrix of input FMs (for the FC layer, there is only one $\mathbf{X}$ vector), $\mathbf{W}_j$ is the $j$-th column of the weight matrix $W$, and $\varphi(\cdot)$ is the neuronal transformation (e.g., ReLU function, here we abandon bias). Now, according to equation (1), the preservation of the activation is equivalent to preserve the inner product.

We introduce a dimension-reduction lemma, named Johnson-Lindenstrauss Lemma (JLL) (Johnson & Lindenstrauss, 1984), to implement the dimension-reduction search with inner product preservation. This lemma states that a set of points in a high-dimensional space can be embedded into a low-dimensional space in such a way that the Euclidean distances between these points are nearly preserved. Specifically, given $0 < \epsilon < 1$, a set of $N$ points in $\mathbb{R}^d$ (i.e., all $\mathbf{X}_i$ and $\mathbf{W}_j$), and a number of $k > O(\frac{log(N)}{\epsilon^2})$, there exists a linear map $f : \mathbb{R}^d \Rightarrow \mathbb{R}^k$ such that

$$(1 - \epsilon)\|\mathbf{X}_i - \mathbf{W}_j\|^2 \le \|f(\mathbf{X}_i) - f(\mathbf{W}_j)\|^2 \le (1 + \epsilon)\|\mathbf{X}_i - \mathbf{W}_j\|^2 \tag{2}$$

for any given $\mathbf{X}_i$ and $\mathbf{W}_j$ pair, where $\epsilon$ is a hyper-parameter to control the approximation error, i.e., larger $\epsilon \Rightarrow$ larger error. When $\epsilon$ is sufficiently small, one corollary from JLL is the following norm preservation (Vu, 2016; Kakade & Shakhnarovich, 2009):

$$P[\, (1 - \epsilon)\|\mathbf{Z}\|^2 \le \|f(\mathbf{Z})\|^2 \le (1 + \epsilon)\|\mathbf{Z}\|^2 \,] \ge 1 - O(\epsilon^2) \tag{3}$$

where $\mathbf{Z}$ could be any $\mathbf{X}_i$ or $\mathbf{W}_j$, and $P$ denotes a probability. It means the vector norm can be preserved with a high probability controlled by $\epsilon$. Given these basics, we can further get the inner product preservation:

$$P[\,|\langle f(\mathbf{X}_i), f(\mathbf{W}_j)\rangle - \langle \mathbf{X}_i, \mathbf{W}_j\rangle| \le \epsilon\,] \ge 1 - O(\epsilon^2). \tag{4}$$

The detailed proof can be found in Appendix A.

Random projection (Vu, 2016; Ailon & Chazelle, 2009; Achlioptas, 2001) is widely used to construct the linear map $f(\cdot)$. Specifically, the original $d$-dimensional vector is projected to a $k$-dimensional ($k \ll d$) one, using a random $k \times d$ matrix $\mathbf{R}$. Then we can reduce the dimension of all $\mathbf{X}_i$ and $\mathbf{W}_j$ by

$$f(\mathbf{X}_i) = \frac{1}{\sqrt{k}}\mathbf{R}\mathbf{X}_i \in \mathbb{R}^k, \;\; f(\mathbf{W}_j) = \frac{1}{\sqrt{k}}\mathbf{R}\mathbf{W}_j \in \mathbb{R}^k. \tag{5}$$

The random projection matrix $\mathbf{R}$ can be generated from Gaussian distribution (Ailon & Chazelle, 2009). In this paper, we adopt a simplified version, termed as sparse random projection (Achlioptas, 2001; Bingham & Mannila, 2001; Li et al., 2006) with

$$P(\mathbf{R}_{pq} = \sqrt{s}) = \frac{1}{2s}; \;\; P(\mathbf{R}_{pq} = 0) = 1 - \frac{1}{s}; \;\; P(\mathbf{R}_{pq} = -\sqrt{s}) = \frac{1}{2s} \tag{6}$$

for all elements in $\mathbf{R}$. This $\mathbf{R}$ only has ternary values that can remove the multiplications during projection, and the remained additions are very sparse. Therefore, the projection overhead is negligible compared to other high-precision operations involving multiplication. Here we set $s = 3$ with 67% sparsity in statistics.

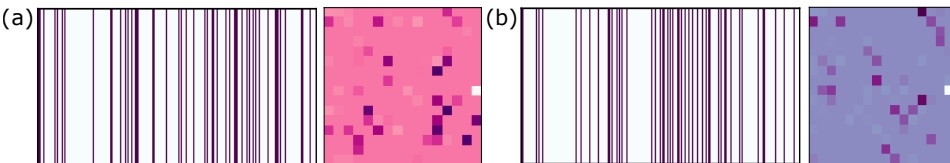

Figure 4: Structured selection via dynamic dimension-reduction search for producing sparse pattern of neuronal activations.

Equation (4) indicates the low-dimensional inner product $\langle f(\mathbf{X}_i), f(\mathbf{W}_j)\rangle$ can still approximate the original high-dimensional one $\langle \mathbf{X}_i, \mathbf{W}_j\rangle$ in equation (1) if the reduced dimension is sufficiently high. Therefore, it is possible to calculate equation (1) in a low-dimensional space for activation estimation, and select the important neurons. As shown in Figure 3(b), each sliding window dynamically selects its own important neurons for the calculation in high-dimensional space, marked in red and blue as two examples. Figure 4 visualizes two sliding windows in a real network to help understand the dynamic process of dimension-reduction search. Here the neuronal activation vector ($n_K$ length) is reshaped to a matrix for clarity. Now For the CONV layer, the computational complexity is only $O[\,m \times n_{PQ} \times n_K \times (k + (1 - \gamma) \times n_{CRS})\,]$, which is less than the original high-dimensional computation with $O(m \times n_{PQ} \times n_{CRS} \times n_K)$ complexity because we usually have $[\,k + (1 - \gamma) \times n_{CRS}\,] \ll n_{CRS}$. For the FC layer, we also have $O[\,m \times n_K \times (k + (1 - \gamma) \times n_C)\,] \ll O(m \times n_C \times n_K)$.

## 2.3 DOUBLE-MASK SELECTION FOR BN COMPATIBILITY

To deal with the important but intractable BN layer, we propose a double-mask selection method presented in Figure 2(c). After the dimension-reduction search based importance estimation, we produce a sparsifying mask that removes the unimportant neurons. The ReLU activation function can maintain this mask by inhibiting the negative activation (actually all the activations of the CONV layer or FC layer after the selection mask are positive with reasonably large sparsity). However, the BN layer will damage this sparsity through inter-sample activation fusion. To address this issue, we copy the same selection mask before the BN layer and directly use it on the BN output. It is straightforward but reasonable because we find that although BN causes the zero activation to be non-zero (Figure 1(f)), these non-zero activations are still very small and can also be removed. This is because BN just scales and shifts the activations that won't change the relative sort order. In this way, we can achieve fully sparse activation dataflow. The back propagated gradients will also be forcibly sparsified every time they pass a mask layer.

## 3 EXPERIMENTAL RESULTS

### 3.1 EXPERIMENT SETUP

The overall training algorithm is presented in Appendices B. Going through the dataflow where the red color denotes the sparse tensors, a widespread sparsity in both the forward and backward passes is demonstrated. The projection matrices are fixed after a random initialization at the beginning of training. We just update the projected weights in the low-dimensional space every 50 iterations to reduce the projection overhead. The detailed search method and the computational complexity of the dimension-reduction search are provided in Appendix B. Regarding the evaluation network models, we use LeNet (LeCun et al., 1998) and a multi-layered perceptron (MLP) on small-scale FASHION dataset (Xiao et al., 2017), VGG8 (Courbariaux et al., 2016; Deng et al., 2018)/ResNet8 (a customized ResNet-variant with 3 residual blocks and 2 FC layers)/ResNet20/WRN-8-2 (Zagoruyko & Komodakis, 2016) on medium-scale CIFAR10 dataset (Krizhevsky & Hinton, 2009), VGG8/WRN-8-2 on another medium-scale CIFAR100 dataset (Krizhevsky & Hinton, 2009), and AlexNet (Krizhevsky et al., 2012)/VGG16 (Simonyan & Zisserman, 2014)/ResNet18, ResNet152 (He et al., 2016)/WRN-18-2 (Zagoruyko & Komodakis, 2016) on large-scale ImageNet dataset (Deng et al., 2009) as workloads. The programming framework is PyTorch and the training platform is based on NVIDIA Titan Xp GPU. We adopt the zero-value compression method (Zhang et al., 2000; Vijaykumar et al., 2015; Rhu et al., 2018) for memory compression and MKL compute library (Wang et al., 2014) on Intel Xeon CPU for acceleration evaluation.

### 3.2 ACCURACY ANALYSIS

In this section, we provide a comprehensive analysis regarding the influence of sparsity on accuracy and explore the robustness of MLP and CNN, the graph selection strategy, the BN compatibility, and the importance of width and depth.

**Accuracy using DSG.** Figure 5(a) presents the accuracy curves on small and medium scale models by using DSG under different sparsity levels. Three conclusions are observed: 1) The proposed DSG affects little on the accuracy when the sparsity is <60%, and the accuracy will present an abrupt descent with sparsity larger than 80%. 2) Usually, the ResNet model family is more sensitive to the sparsity increasing due to fewer parameters than the VGG family. For the VGG8 on CIFAR10, the accuracy loss is still within 0.5% when sparsity reaches 80%. 3) Compared to MLP, CNN can tolerate more sparsity. Figure 5(b) further shows the results on large scale models on ImageNet. Because training large model is time costly, we only present several experimental points. Consistently, the VGG16 shows better robustness compared to the ResNet18, and the WRN with wider channels on each layer performs much better than the other two models. We will discuss the topic of width and depth later.

**Graph Selection Strategy.** To investigate the influence of graph selection strategy, we repeat the sparsity vs. accuracy experiments on CIFAR10 under different selection methods. Two baselines are used here: the oracle one that keeps the neurons with top-k activations after the whole VMM computation at each layer, and the random one that randomly selects neurons to keep. The results are shown in Figure 5(c), in which we can see that our dimension-reduction search and the oracle one perform much better than the random selection under high sparsity condition. Moreover, dimension-reduction search achieves nearly the same accuracy with the oracle top-k selection, which indicates the proposed random projection method can find an accurate activation estimation in the low-dimensional space. In detail, Figure 5(d) shows the influence of parameter $\epsilon$ that reflects the degree of dimension reduction. Lower $\epsilon$ can approach the original inner product more accurately, that brings higher accuracy but at the cost of more computation for graph selection since less dimension reduction. With $\epsilon = 0.5$, the accuracy loss is within 1% even if the sparsity reaches 80%.

**BN Compatibility.** Figure 5(e) focuses the BN compatibility issue. Here we use dimension-reduction search for the graph sparsifying, and compare three cases: 1) removing the BN operation and using single mask; 2) keeping BN and using only single mask (the first one in Figure 2(c)); 3) keeping BN and using double masks (i.e. double-mask selection). The one without BN is very sensitive to the graph ablation, which indicates the importance of BN for training. Comparing the two with BN, the double-mask selection even achieves better accuracy since the regularization effect.

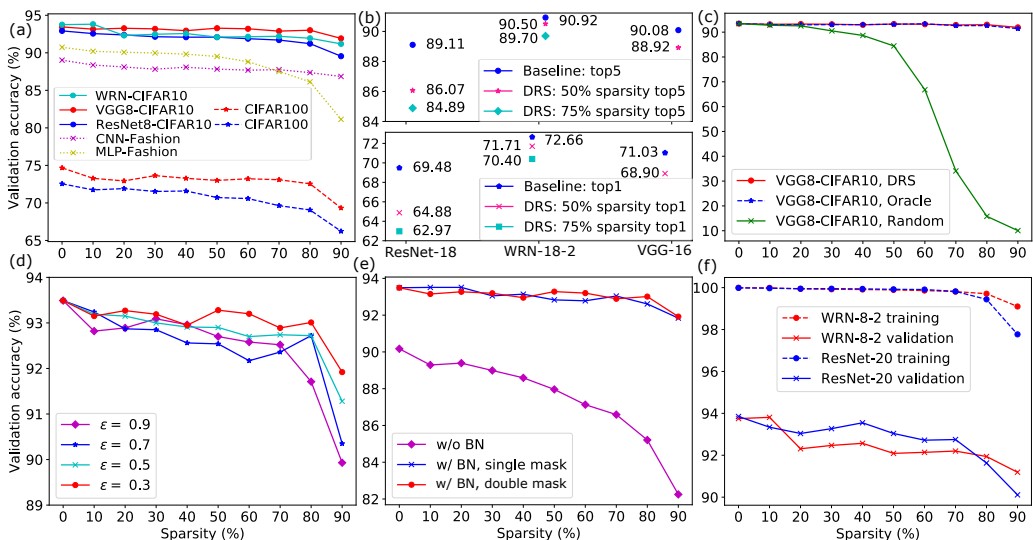

Figure 5: Comprehensive analysis on sparsity v.s. accuracy. (a) & (b) Accuracy using DSG; (c) Influence of the graph selection strategy; (d) Influence of the dimension-reduction degree; (e) Influence of the double-mask selection for BN compatibility; (f) Influence of the network depth and width. 'DRS' denotes dimension-reduction search.

This observation indicates the effectiveness of the proposed double-mask selection for simultaneously recovering the sparsity damaged by the BN layer and maintaining the accuracy.

**Width or Depth.** Furthermore, we investigate an interesting comparison regarding the network width and depth, as shown in Figure 5(f). On the training set, WRN with fewer but wider layers demonstrates more robustness than the deeper one with more but slimmer layers. On the validation set, the results are a little more complicated. Under small and medium sparsity, the deeper ResNet performs better (1%) than the wider one. While when the sparsity increases substantial (>75%), WRN can maintain the accuracy better. This indicates that, in medium-sparse space, the deeper network has stronger representation ability because of the deep structure; however, in ultra-high-sparse space, the deeper structure is more likely to collapse since the accumulation of the pruning error layer by layer. In reality, we can determine which type of model to use according to the sparsity requirement. In Figure 5(b) on ImageNet, the reason why WRN-18-2 performs much better is that it has wider layers without reducing the depth.

**Convergence.** DSG does not slow down the convergence speed, which can be seen from Figure 10(a)-(b) in Appendix C. This owes to the high fidelity of inner product when we use random projection to reduce the data dimension, as shown in Figure 10(c). Interestingly, Figure 11 (also in Appendix C) reveals that the selection mask for each sample also converges as training goes on, however, the selection pattern varies across samples. To save the selection patterns of all samples is memory consuming, which is the reason why we do not directly suspend the selection patterns after training but still do on-the-fly dimension-reduction search in inference.

## 3.3 REPRESENTATIONAL COST REDUCTION

This section presents the benefits from DSG on representational cost. We measure the memory consumption over five CNN benchmarks on both the training and inference phases. For data compression, we use zero-value compression algorithm (Zhang et al., 2000; Vijaykumar et al., 2015; Rhu et al., 2018). Figure 6 shows the memory optimization results, where the model name, mini-batch size, and the sparsity are provided. In training, besides the parameters, the activations across all layers should be stashed for the backward computation. Consistent with the observation mentioned above that the neuron activation beats weight to dominate memory overhead, which is different from the previous work on inference. We can reduce the overall representational cost by average 1.7x (2.72 GB), 3.2x (4.51 GB), and 4.2x (5.04 GB) under 50%, 80% and 90% sparsity, respec-

tively. If only considering the neuronal activation, these ratios could be higher up to 7.1x. The memory overhead for the selection masks is minimal (<2%).

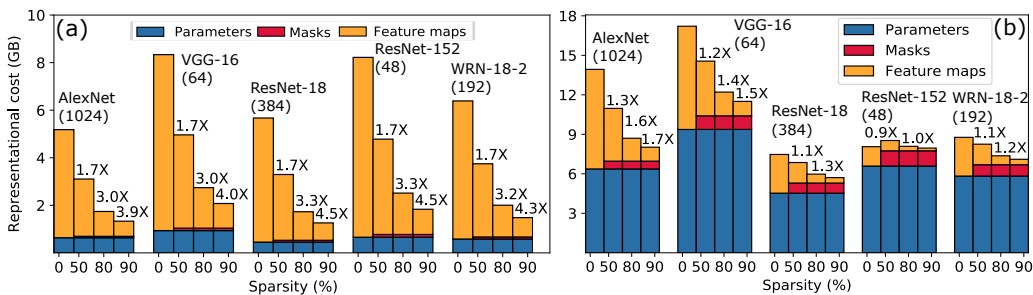

Figure 6: Memory footprint comparisons for (a) training and (b) inference.

During inference, only memory space to store the parameters and the activations of the layer with maximum neuron amount is required. The benefits in inference are relatively smaller than that in training since weight is the dominant memory. On ResNet152, the extra mask overhead even offsets the compression benefit under 50% sparsity, whereas, we can still achieve up to 7.1x memory reduction for activations and 1.7x for overall memory. Although the compression is limited for inference, it still can achieve noticeable acceleration that will be shown in the next section. Moreover, reducing costs for both training and inference is our major contribution.

## 3.4    COMPUTATIONAL COST REDUCTION

We assess the results on reducing the computational cost for both training and inference. As shown in Figure 7, both the forward and backward pass consume much fewer operations, i.e., multiply-and-accumulate (MAC). On average, 1.4x (5.52 GMACs), 1.7x (9.43 GMACs), and 2.2x (10.74 GMACs) operation reduction are achieved in training under 50%, 80% and 90% sparsity, respectively. For inference with only forward pass, the results increase to 1.5x (2.26 GMACs), 2.8x (4.22 GMACs), and 3.9x (4.87 GMACs), respectively. The overhead of the dimension-reduction search in the low-dimensional space is relatively larger (<6.5% in training and <19.5% in inference) compared to the mask overhead in memory cost. Note that the training demonstrates less improvement than the inference, which is because the acceleration of the backward pass is partial. The error propagation is accelerative, but the weight gradient generation is not because of the irregular sparsity that is hard to obtain practical acceleration. Although the computation of this part is also very sparse with much fewer operations [2], we do not include its GMACs reduction for practical concern.

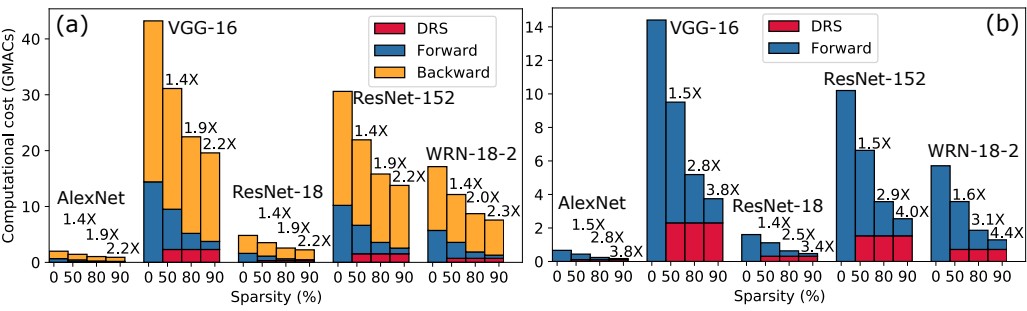

Figure 7: Computational complexity comparisons for (a) training and (b) inference. 'DRS' denotes dimension-reduction search.

Finally, we evaluate the execution time on CPU using Intel MKL kernels (Wang et al. (2014)). As shown in Figure 8(a), we evaluate the execution time of these layers after the dimension-reduction search on VGG8. Comparing to VMM baselines, our approach can achieve 2.0x, 5.0x, and 8.5x average speedup under 50%, 80%, and 90% sparsity, respectively. When the baselines change to

---
[2]See Algorithm 1 in Appendices B

GEMM (general matrix multiplication), the average speedup decreases to 0.6x, 1.6x, and 2.7x, respectively. The reason is that DSG generates dynamic vector-wise sparsity, which is not well supported by GEMM. A potential way to improve GEMM-based implementation, at workload mapping and tiling time, is reordering executions at the granularity of vector inner-product and grouping non-redundant executions to the same tile to improve local data reuse.

On the same network, we further compare our approach with smaller dense models which could be another way to reduce the computational cost. As shown in Figure 8(b), comparing with dense baseline, our approach can reduce training time with little accuracy loss. Even though the equivalent smaller dense models with the same effective nodes, i.e., reduced MACs, save more training time, the accuracy is much worse than our DSG approach. Figure 12 in Appendix D gives more results on ResNet8 and AlexNet.

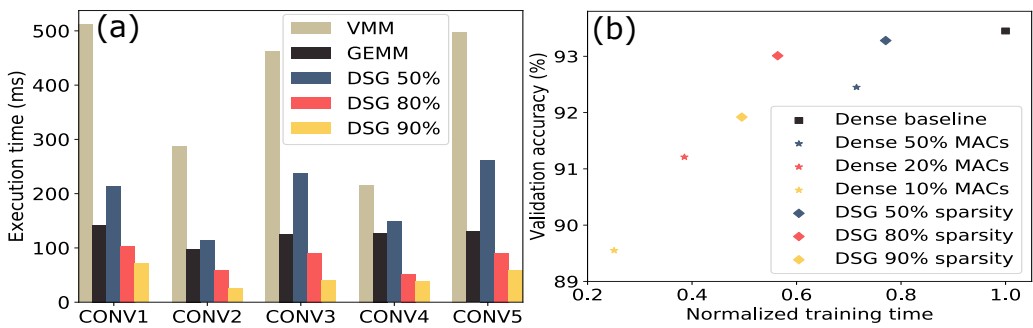

Figure 8: On VGG8: (a) Layer-wise execution time comparison; (b) Validation accuracy v.s. training time of different models: large-sparse ones and smaller-dense ones with equivalent MACs.

## 4 RELATED WORK

**DNN Compression** (Ardakani et al., 2016) achieved up to 90% weight sparsity by randomly removing connections. (Han et al., 2015b;a) reduced the weight parameters by pruning the unimportant connections. The compression is mainly achieved on FC layers, which makes it ineffective for CONV layer-dominant networks, e.g., ResNet. To improve the pruning performance, Y. He et al. (He et al., 2018b) leveraged reinforcement learning to optimize the sparsity configuration across layers. However, it is difficult to obtain practical speedup due to the irregularity of the element-wise sparsity (Han et al., 2015b;a). Even if designing ASIC from scratch (Han et al., 2016; 2017), the index overhead is enormous and it only works under high sparsity. These methods usually require a pre-trained model, iterative pruning, and fine-tune retraining, that targets inference optimization.

**DNN Acceleration** Different from compression, the acceleration work consider more on the sparse pattern. In contrast to the fine-grain compression, coarse-grain sparsity was further proposed to optimize the execution speed. Channel-level sparsity was gained by removing unimportant weight filters (He et al., 2018a; Chin et al., 2018), training penalty coefficients (Liu et al., 2017; Ye et al., 2018; Luo & Wu, 2018), or solving optimization problem (Luo et al., 2017; He et al., 2017; Liang et al., 2018; Hu et al., 2018). Wen et al. (2016) introduced a L2-norm group-lasso optimization for both medium-grain sparsity (row/column) and coarse-grain weight sparsity (channel/filter/layer). Molchanov et al. (2016) introduced the Taylor expansion for neuron pruning. However, they just benefit the inference acceleration, and the extra solving of the optimization problem usually makes the training more complicated. Lin et al. (2017a) demonstrated predicting important neurons then bypassed the unimportant ones via low-precision pre-computation with less cost. Spring & Shrivastava (2017) leveraged the randomized hashing to predict the important neurons. However, the hashing search aims at finding neurons whose weight bases are similar to the input vector, which cannot estimate the inner product accurately thus will probably cause significant accuracy loss on large models. Sun et al. (2017) used a straightforward top-k pruning on the back propagated errors for training acceleration. But they only simplified the backward pass and presented the results on tiny FC models. Furthermore, the BN compatibility problem that is very important for large-model training still remains untouched. Lin et al. (2017b) pruned the gradients for accelerating distributed

training, but the focus is on multi-node communication rather than the single-node scenario discussed in this paper.

## 5 CONCLUSION

In this work, we propose DSG (dynamic and sparse graph) structure for efficient DNN training and inference through a dimension-reduction search based sparsity forecast for compressive memory and accelerative execution and a double-mask selection for BN compatibility without sacrificing model's expressive power. It can be easily extended to the inference by using the same selection pattern after training. Our experiments over various benchmarks demonstrate significant memory saving (up to 4.5x for training and 1.7x for inference) and computation reduction (up to 2.3x for training and 4.4x for inference). Through significantly boosting both forward and backward passes in training, as well as in inference, DSG promises efficient deep learning in both the cloud and edge.

## ACKNOWLEDGMENT

This work was partially supported by the National Science Foundations(NSF) under Grant No. 1725447 and 1730309, the National Natural Science Foundation of China under Grant No. 61603209 and 61876215. Financial support from the Beijing Innovation Center for Future Chip is also gratefully acknowledged.

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

## APPENDIX A   PROOF OF THE DIMENSION-REDUCTION SEARCH FOR INNER PRODUCT PRESERVATION

**Theorem 1.** Given a set of $N$ points in $\mathbb{R}^d$ (i.e. all $\mathbf{X}_i$ and $\mathbf{W}_j$), and a number of $k > O(\frac{log(N)}{\epsilon^2})$, there exist a linear map $f : \mathbb{R}^d \Rightarrow \mathbb{R}^k$ and a $\epsilon_0 \in (0, 1)$, for $0 < \epsilon \leq \epsilon_0$ we have

$$P[\ |\langle f(\mathbf{X}_i), f(\mathbf{W}_j)\rangle - \langle \mathbf{X}_i, \mathbf{W}_j\rangle| \leq \epsilon\ ] \geq 1 - O(\epsilon^2). \tag{7}$$

for all $\mathbf{X}_i$ and $\mathbf{W}_j$.

**Proof**. According to the definition of inner product and vector norm, any two vectors $\mathbf{a}$ and $\mathbf{b}$ satisfy

$$\begin{cases} \langle \mathbf{a}, \mathbf{b}\rangle = (\|\mathbf{a}\|^2 + \|\mathbf{b}\|^2 - \|\mathbf{a} - \mathbf{b}\|^2)/2 \\ \langle \mathbf{a}, \mathbf{b}\rangle = (\|\mathbf{a} + \mathbf{b}\|^2 - \|\mathbf{a}\|^2 - \|\mathbf{b}\|^2)/2 \end{cases}. \tag{8}$$

It is easy to further get

$$\langle \mathbf{a}, \mathbf{b}\rangle = (\|\mathbf{a} + \mathbf{b}\|^2 - \|\mathbf{a} - \mathbf{b}\|^2)/4. \tag{9}$$

Therefore, we can transform the target in equation (7) to

$$\begin{aligned} &|\ \langle f(\mathbf{X}_i), f(\mathbf{W}_j)\rangle - \langle \mathbf{X}_i, \mathbf{W}_j\rangle\ | \\ =\ &|\ \|f(\mathbf{X}_i) + f(\mathbf{W}_j)\|^2 - \|f(\mathbf{X}_i) - f(\mathbf{W}_j)\|^2 - \|\mathbf{X}_i + \mathbf{W}_j\|^2 + \|\mathbf{X}_i - \mathbf{W}_j\|^2\ |/4 \\ \leq\ &|\ \|f(\mathbf{X}_i) + f(\mathbf{W}_j)\|^2 - \|\mathbf{X}_i + \mathbf{W}_j\|^2\ |/4 + |\ \|f(\mathbf{X}_i) - f(\mathbf{W}_j)\|^2 - \|\mathbf{X}_i - \mathbf{W}_j\|^2\ |/4, \end{aligned} \tag{10}$$

which is also based on the fact that $|u - v| \leq |u| + |v|$. Now recall the definition of random projection in equation (5) of the main text

$$f(\mathbf{X}_i) = \frac{1}{\sqrt{k}}\mathbf{R}\mathbf{X}_i \in \mathbb{R}^k, \quad f(\mathbf{W}_j) = \frac{1}{\sqrt{k}}\mathbf{R}\mathbf{W}_j \in \mathbb{R}^k. \tag{11}$$

Substituting equation (11) into equation (10), we have

$$\begin{aligned} &|\ \langle f(\mathbf{X}_i), f(\mathbf{W}_j)\rangle - \langle \mathbf{X}_i, \mathbf{W}_j\rangle\ | \\ \leq\ &|\ \|\tfrac{1}{\sqrt{k}}\mathbf{R}\mathbf{X}_i + \tfrac{1}{\sqrt{k}}\mathbf{R}\mathbf{W}_j\|^2 - \|\mathbf{X}_i + \mathbf{W}_j\|^2\ |/4 + |\ \|\tfrac{1}{\sqrt{k}}\mathbf{R}\mathbf{X}_i - \tfrac{1}{\sqrt{k}}\mathbf{R}\mathbf{W}_j\|^2 - \|\mathbf{X}_i - \mathbf{W}_j\|^2\ |/4 \\ =\ &|\ \|\tfrac{1}{\sqrt{k}}\mathbf{R}(\mathbf{X}_i + \mathbf{W}_j)\|^2 - \|\mathbf{X}_i + \mathbf{W}_j\|^2\ |/4 + |\ \|\tfrac{1}{\sqrt{k}}\mathbf{R}(\mathbf{X}_i - \mathbf{W}_j)\|^2 - \|\mathbf{X}_i - \mathbf{W}_j\|^2\ |/4 \\ =\ &|\ \|f(\mathbf{X}_i + \mathbf{W}_j)\|^2 - \|\mathbf{X}_i + \mathbf{W}_j\|^2\ |/4 + |\ \|f(\mathbf{X}_i - \mathbf{W}_j)\|^2 - \|\mathbf{X}_i - \mathbf{W}_j\|^2\ |/4 \end{aligned} \tag{12}$$

Further recalling the norm preservation in equation (3) of the main text: there exist a linear map $f : \mathbb{R}^d \Rightarrow \mathbb{R}^k$ and a $\epsilon_0 \in (0, 1)$, for $0 < \epsilon \leq \epsilon_0$ we have

$$P[\ (1 - \epsilon)\|\mathbf{Z}\|^2 \leq \|f(\mathbf{Z})\|^2 \leq (1 + \epsilon)\|\mathbf{Z}\|^2\ ] \geq 1 - O(\epsilon^2). \tag{13}$$

Substituting the equation (13) into equation (12) yields

$$\begin{aligned} P[\ &|\ \|f(\mathbf{X}_i + \mathbf{W}_j)\|^2 - \|\mathbf{X}_i + \mathbf{W}_j\|^2\ |/4 + |\ \|f(\mathbf{X}_i - \mathbf{W}_j)\|^2 - \|\mathbf{X}_i - \mathbf{W}_j\|^2\ |/4... \\ &\leq \tfrac{\epsilon}{4}(\|\mathbf{X}_i + \mathbf{W}_j\|^2 + \|\mathbf{X}_i - \mathbf{W}_j\|^2) = \tfrac{\epsilon}{2}(\|\mathbf{X}_i\|^2 + \|\mathbf{W}_j\|^2)\ ]... \\ \geq P(\ &|\ \|f(\mathbf{X}_i + \mathbf{W}_j)\|^2 - \|\mathbf{X}_i + \mathbf{W}_j\|^2\ |/4 \leq \tfrac{\epsilon}{4}\|\mathbf{X}_i + \mathbf{W}_j\|^2\ )... \\ \times P(\ &|\ \|f(\mathbf{X}_i - \mathbf{W}_j)\|^2 - \|\mathbf{X}_i - \mathbf{W}_j\|^2\ |/4 \leq \tfrac{\epsilon}{4}\|\mathbf{X}_i - \mathbf{W}_j\|^2\ )... \\ \geq\ &[1 - O(\epsilon^2)] \cdot [1 - O(\epsilon^2)] = 1 - O(\epsilon^2). \end{aligned} \tag{14}$$

Combining equation (12) and (14), finally we have

$$P[\ |\ \langle f(\mathbf{X}_i), f(\mathbf{W}_j)\rangle - \langle \mathbf{X}_i, \mathbf{W}_j\rangle\ | \leq \tfrac{\epsilon}{2}(\|\mathbf{X}_i\|^2 + \|\mathbf{W}_j\|^2)\ ] \geq 1 - O(\epsilon^2). \tag{15}$$

It can be seen that, for any given $\mathbf{X}_i$ and $\mathbf{W}_j$ pair, the inner product can be preserved if the $\epsilon$ is sufficiently small. Actually, previous work (Achlioptas, 2001; Bingham & Mannila, 2001; Vu, 2016) discussed a lot on the random projection for various big data applications, here we re-organize these supporting materials to form a systematic proof. We hope this could help readers to follow this paper. In practical experiments, there exists a trade-off between the dimension reduction degree and the recognition accuracy. Smaller $\epsilon$ usually brings more accurate inner product estimation and better recognition accuracy while at the cost of higher computational complexity with larger $k$, and vice versa. Because the $\|\mathbf{X}_i\|^2$ and $\|\mathbf{W}_j\|^2$ are not strictly bounded, the approximation may suffer from some noises. Anyway, from the abundant experiments in this work, the effectiveness of our approach for training dynamic and sparse neural networks has been validated.

**Data:** A mini-batch of inputs & targets $(\mathbf{X}_0, \mathbf{X}^*)$, previous weights $\mathbf{W}^t$, previous BN parameters $\theta^t$.
**Result:** Update weights $\mathbf{W}^{t+1}$, update BN parameters $\theta^{t+1}$.

Random projection: $f(\mathbf{W}_k^t) \Leftarrow \mathbf{W}_k^t$;

Step 1. Forward Computation;
**for** *k=1 to L* **do**
    **if** *k<L* **then**
        Projection: $f(\mathbf{X}_{k-1}) \Leftarrow \mathbf{X}_{k-1}$;
        Generating $Mask_k$ via dimension-reduction search according to $f(\mathbf{X}_{k-1})$ and $f(\mathbf{W}_k^t)$;
        $\mathbf{S}_k \Leftarrow \varphi[\, Mask_k(\mathbf{X}_{k-1}\mathbf{W}_k^t)\,]$;
        $\mathbf{X}_k \Leftarrow Mask_k[\, BN(\mathbf{S}_k, \theta_k^t)\,]$;
    **else**
        $\mathbf{X}_L \Leftarrow linear(\mathbf{X}_{L-1}\mathbf{W}_L^t)$;
    **end**
**end**

Step 2. Backward Computation;
Compute the gradient of the output layer $\mathbf{G}_{\mathbf{X}_L} = \frac{\partial C(\mathbf{X}_L, \mathbf{X}^*)}{\partial \mathbf{X}_L}$;
**for** *k=L to 1* **do**
    **if** *k==L* **then**
        $\mathbf{G}_{\mathbf{X}_{L-1}} \Leftarrow Mask_{k-1}(\mathbf{G}_{\mathbf{X}_L}(\mathbf{W}_L^t)^T)$;
        $\mathbf{G}_{\mathbf{W}_L} \Leftarrow \mathbf{G}_{\mathbf{X}_L}^T \mathbf{X}_{L-1}$;
    **else**
        $(\mathbf{G}_{\mathbf{S}_k}, \mathbf{G}_{\theta_k}) \Leftarrow Mask_k[\, BN\_grad(\mathbf{G}_{\mathbf{X}_k}, \mathbf{S}_k, \theta_k^t)\,]$;
        $\mathbf{G}_{\mathbf{W}_k} \Leftarrow (\mathbf{G}_{\mathbf{S}_k} \odot \varphi\_grad)^T \mathbf{X}_{k-1}$;
        **if** *k>1* **then**
            $\mathbf{G}_{\mathbf{X}_{k-1}} \Leftarrow Mask_{k-1}[\, (\mathbf{G}_{\mathbf{S}_k} \odot \varphi\_grad)(\mathbf{W}_k^t)^T\,]$;
        **end**
    **end**
**end**

Step 3. Parameter Update;
**for** *k=1 to L* **do**
    $\mathbf{W}_k^{t+1} \Leftarrow Optimizer(\mathbf{W}_k^t, \mathbf{G}_{\mathbf{W}_k})$;
    $\theta_k^{t+1} \Leftarrow Optimizer(\theta_k^t, \mathbf{G}_{\theta_k})$;
**end**

**Algorithm 1:** DSG training

## APPENDIX B    IMPLEMENTATION AND OVERHEAD

The training algorithm for generating DSG is presented in Algorithm 1. The generation procedure of the critical neuron mask based on the virtual activations estimated in the low-dimensional space is presented in Figure 9, which is a typical top-$k$ search. The $k$ value is determined by the activation size and the desired sparsity $\gamma$. To reduce the search cost, we calculate the first input sample $X(1)$ within the current mini-batch and then conduct a top-$k$ search over the whole virtual activation matrix for obtaining the top-$k$ threshold under this sample. The remaining samples share the top-$k$ threshold from the first sample to avoid costly searching overhead. At last, the overall activation mask is generated by setting the mask element to one if the estimated activation is larger than the top-$k$ threshold and setting others to zero. In this way, we greatly reduce the search cost. Note that, for the FC layer, each sample $X(i)$ is a vector.

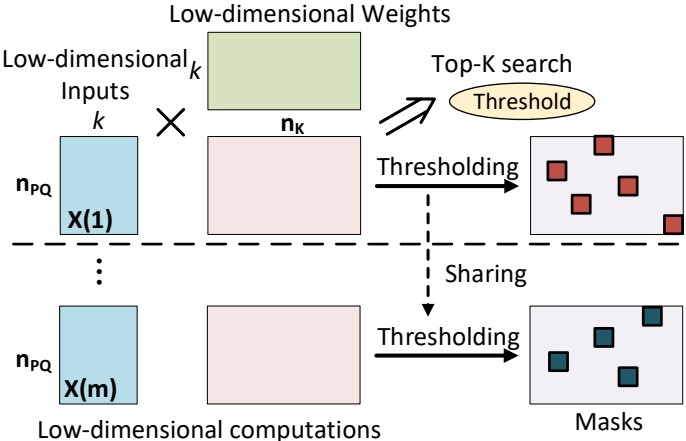

Figure 9: Selection mask generation: using a top-$k$ search on the first input sample $X(1)$ within each mini-batch to obtain a top-$k$ threshold which is shared by the following samples. Then, we apply thresholding on the whole output activation tensor to generate the importance mask for the same mini-batch.

Table 1: Computational complexity of dimension-reduction search. MMACs denotes mega-MACs and BL denotes baseline.

| Layers | Dimension | | | | | Operations (MMACs) | | | | |
|---|---|---|---|---|---|---|---|---|---|---|
| $n_{PQ}, n_{CRS}, n_K$ | BL | 0.3 | 0.5 | 0.7 | 0.9 | BL | 0.3 | 0.5 | 0.7 | 0.9 |
| 1024, 1152, 128 | 1152 | 539 | 232 | 148 | 119 | 144 | 67.37 | 29 | 18.5 | 14.88 |
| 256, 1152, 256 | 1152 | 616 | 266 | 169 | 136 | 72 | 38.5 | 16.63 | 10.56 | 8.5 |
| 256, 2304, 256 | 2304 | 616 | 266 | 169 | 136 | 144 | 38.5 | 16.63 | 10.56 | 8.5 |
| 64, 2304, 512 | 2304 | 693 | 299 | 190 | 154 | 72 | 21.65 | 9.34 | 5.94 | 4.81 |
| 64, 4608, 512 | 4608 | 693 | 299 | 190 | 154 | 144 | 21.65 | 9.34 | 5.94 | 4.81 |

Furthermore, we investigate the influence of the $\epsilon$ on the computation cost of dimension-reduction search for importance estimation. We take several layers from the VGG8 on CIFAR10 as a case study, as shown in Table 1. With $\epsilon$ larger, the dimension-reduction search can achieve lower dimension with much fewer operations. The average reduction of the dimension is 3.6x ($\epsilon = 0.3$), 8.5x ($\epsilon = 0.5$), 13.3x ($\epsilon = 0.7$), and 16.5x ($\epsilon = 0.9$). The resulting operation reduction is 3.1x, 7.1x, 11.1x, and 13.9x, respectively.

## APPENDIX C    CONVERGENCE ANALYSIS

One interesting question is that whether DSG slows down the training convergence or not, which is answered by Figure 10. According to Figure 10(a)-(b), the convergence speed under DSG con-

straints varies little from the vanilla model training. This probably owes to the high fidelity of inner product when we use random projection to reduce the data dimension. Figure 10(c) visualizes the distribution of the pairwise difference between the original high-dimensional inner product and the low-dimensional one for the CONV5 layer of VGG8 on CIFAR10. Most of the inner product differences are around zero, which implies an accurate approximation capability of the proposed dimension-reduction search. This helps reduce the training variance and avoid training deceleration.

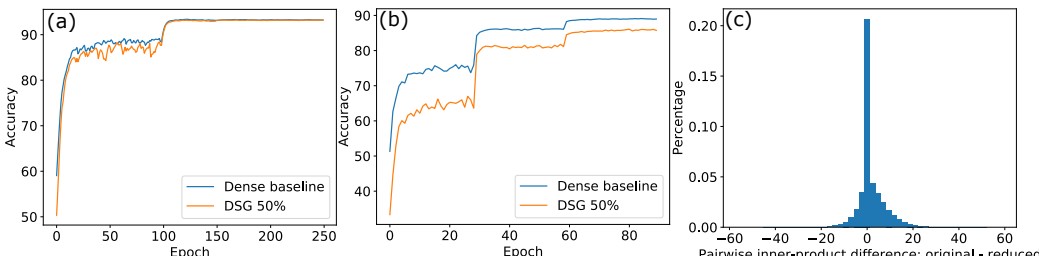

Figure 10: Accuracy convergence. (a) Training curve with validation accuracy of VGG8 on CI-FAR10; (b) Training curve with top-5 validation accuracy of ResNet-18 on ImageNet; (c) Distribution of pairwise difference between the original high-dimensional inner product and the low-dimensional one for the CONV5 layer in VGG8.

Another question in DSG is that whether the selection masks converge during training or not. To explore the answer, we did an additional experiment as shown in the Figure 11. We select a mini-batch of training samples as a case study for data recording. Each curve presents the results of one layer (CONV2-CONV6). For each sample at each layer, we recorded the change of binary selection mask between two adjacent training epochs. Here the change is obtained by calculating the $L1$-norm value of the difference tensor of two mask tensors at two adjacent epochs, i.e., $change = batch\_avg\_L1norm(mask_{i+1} - mask_i)$. Here the $batch\_avg\_L1norm(\cdot)$ indicates the average $L1$-norm value across all samples in one mini-batch. As shown in Figure 11(a), the selection mask for each sample converges as training goes on.

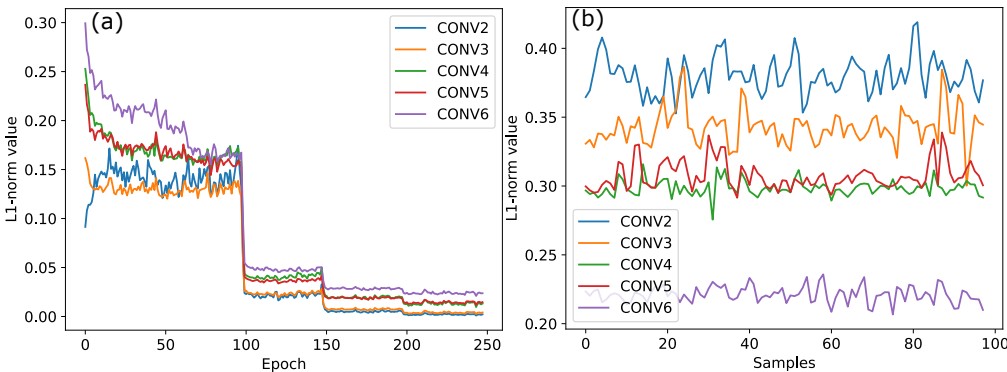

Figure 11: Selection mask convergence. (a) Average $L1$-norm value of the difference mask tensors between adjacent training epochs across all samples in one mini-batch; (b) Average $L1$-norm value of the difference mask tensors between adjacent samples after training.

In our implementation we inherit the random projection matrix from training and perform the same on-the-fly dimension-reduction search in inference. We didn't try to directly suspend the selection masks, because the selection mask might vary across samples even if we observe convergence for each sample. This can be seen from Figure 11(b), where the difference mask tensors between adjacent samples in one mini-batch present significant differences (large $L1$-norm value) after training. Therefore, it will consume lot of memory space to save these trained masks for all samples, which is less efficient than conducting on-the-fly search during inference.

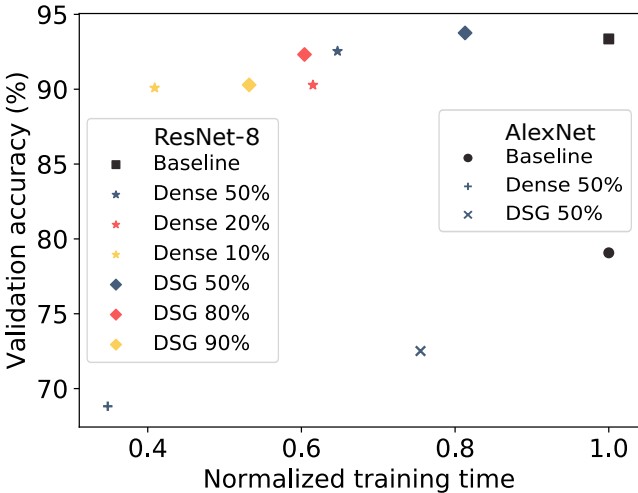

Figure 12: Comparison with smaller-dense models with equivalent MACs using ResNet8 on CI-FAR10 and AlexNet on ImageNet.

## APPENDIX D   COMPARISON WITH OTHER METHODS

Figure 12 extends Figure 8(b) in the main text to more network structures, including ResNet8 on CI-FAR10 and AlexNet on ImageNet. The similar observation can be achieved: the equivalent smaller dense models with the same effective MACs are able to save more training time but the accuracy degradation will be increased. Note that in this figure, the DSG training uses a warm-up training with dense model for the first 10 epochs. The overhead of the warm-up training has been taken account into the entire training cost. To make the accuracy results on CIFAR10 and ImageNet comparable for figure clarity, AlexNet reports the top-5 accuracy.

Our work targets at both the training and inference phases while most of previous work focused on the inference compression. In prior methods, the training usually becomes more complicated with various regularization constraints or iterative fine-tuning/retraining. Therefore, it is not very fair to compare with them during training. For this reason, we just compare with them on the inference pruning. Different from doing DSG training from scratch, here we utilize DSG for fine-tuning based on pre-trained models.

Table 2: Comparison with other structured sparsification methods for inference. All the results are from VGG16 on ImageNet, and the default accuracy is top-1 accuracy. The baseline methods are Taylor Expansion (Molchanov et al., 2016), ThiNet (Luo et al., 2017), Channel Pruning (Hu et al., 2018), AutoPrunner (Luo & Wu, 2018), and AMC (He et al., 2018b).

| Methods | Taylor Expansion | ThiNet | Channel Pruning | AutoPrunner | AMC | **DSG** |
|---|---|---|---|---|---|---|
| Operation Sparsity | 62.86% | 69.81% | 69.32% | 73.6% | 80% | 62.92% |
| Accuracy | 87%(top-5) | 67.34% | 70.42% | 68.43% | 69.1% | 71.44%(top-1) 90.56%(top-5) |

To guarantee the fairness, all the results are from the same network (VGG16) on the same dataset (ImageNet). Since our DSG produces structured sparsity, we also select structured sparsity work as comparison baselines. Different from the previous experiments in this paper, we further take the input sparsity at each layer into account rather than only count the output sparsity. This is due to the fact that the baselines consider all zero operands. The results are listed in Table 2, from which we can see that DSG is able to achieve a good balance between the operation amount and model accuracy.

