# OpenReview forum: "Dynamic Sparse Graph for Efficient Deep Learning"
_ICLR.cc/2019/Conference_

### Official Review · AnonReviewer1 · 2018-10-31
**Beautiful approximation idea. Can it be implemented efficiently?**

**Rating:** 7
**Confidence:** 4

**Review:**

This manuscript introduces a computational method to speed up training and inference in deep neural networks: the method is based on dynamic pruning of the compute graph at each iteration of the SGD to approximate computations with a sparse graph. To select which neurons can be zeros and ignored at a given iteration, the approach computes approximate activations using random projections. The approach gives an overall decrease in run-time of 0.8 to 0.6. I believe that its largest drawback is that it does not lead to the same sparsity pattern in a full minibatch, and hence cannot be implemented using matrix-matrix multiplications (GEMM). As a result, the compute-time speed ups are not huge, though the decrease in memory is important. In my eyes, this is the largest drawback of the manuscript: the total computational speed-up demonstrated is not fully convincing.

The manuscript is overall well written and easy to understand, though I wish that the authors employed less acronyms which forced me to scan back as I kept forgetting what they mean.

The strength of the paper are that the solution proposed (dynamic approximation) is original and sensible. The limitations are that I am not sure that it can give significant speedups because I it is probably hard to implement to use well the hardware.

Questions and comments:

1. Can the strategy contributed be implemented efficiently on GPUs? It would have been nice to have access to some code.

2. Fig 8(b) is the most important figure, as it gives the overall convergence time. Is the "dense baseline" using matrix-vector operations (VMM) or mini-batched matrix-matrix operation (GEMM)?

3. Can the method be adapted to chose a joint sparsity across a mini-batch? This would probably mean worst approximation properties but would enable the use of matrix-matrix operations.

4. It is disappointing that figure 8 is only on VGG8, rather than across multiple architectures.

5. The strategy of zeroing inputs of layers can easily create variance that slows down overall convergence (see Mensh TSP 2018 for an analysis of such scenario). In stochastic optimization, there a various techniques to recover fast convergence. Do the authors think that such scenario is at play here, and that similar variance-reduction methods could bring benefits?

6. I could not find what results backed the numbers in the conclusion: 2.3 speed up for training. Is this compared to VMM implementations? In which case it is not a good baseline. Is this for one iteration? In which case, it is not what matters at the end.

7. Is there a link between drop-out and the contributed method, for instance if the sparsity was chosen fully random? Can the contributed method have a regularizing effect?

---

> ### Author Response · Authors · 2018-11-16
> **Responses to Reviewer#1 (Part I)**
>
> Thanks so much for your positive feedback on our methods and constructive comments on the implementation. We provide the answers as below.
>
> (1)	Q1-3, 6: GEMM Implementation on GPU:
>   First, all the authors agree that your comments on the GEMM implementation are quite pertinent, which evidences your correct understanding of our DSG method and your expertise on GPU implementation. In fact, in the text descriptions for Fig. 8(a) in the previous submission, we have mentioned that “DSG generates dynamic vector-wise sparsity, which is not well supported by GEMM,” which is consistent with your comments “it does not lead to the same sparsity pattern in a full minibatch and hence cannot be implemented using GEMM.”
>
>   Second, we should note that to achieve practical speedup on GPU is difficult. GPU implementation is based on coarse-grain matrix-matrix multiplications. The problem of our current method is that the vector-wise regular sparsity for VMM will be degraded to irregular sparsity if we aggregate the VMMs for different samples to call GEMM due to the inconsistent sparsity pattern for each VMM. To accelerate irregular sparse GEMM on GPU is a well-known hard problem for the NN acceleration community.
>
>   This is the reason why many of prior work cannot achieve practical GPU speedup even if they claimed high compression ratio. This is also the reason why many researches study structured sparsity. However, more structured sparsity will compromise more accuracy. Therefore, there is a trade-off between the regularity and accuracy. Furthermore, most of these methods for structured sparsity are for inference compression which usually makes the training more complicated due to various regularization constraints and iterative fine-tuning/retraining. In contrast, we aim at compressing and accelerating both training and inference.
>
>   As the first step, our major goal in this paper is to validate the functionality of the proposed DSG method based on dimension reduction. Although our sparsity pattern is not well structured for GEMM, it is still much more regular (vector-wise for each VMM) than the fully irregular sparsity. At this stage, we implement DSG based on multi-thread VMMs. It can do much better than dense VMM while can only outperform GEMM in the case of quite high sparsity (see Fig. 8(a)). Your question of how to further optimize GEMM under the DSG framework is truly interesting and deserves our future investigation.
>
>   Third, although current DSG is not very compatible with GEMM on GPUs, we still have great potential for implementing on specialized hardware. The vector-wise sparsity pattern of DSG is quite regular for VMM. Moreover, many specialized NN accelerators are based on VMM operations that are suitable for using our DSG method. With customized hardware design, it is easy to achieve high speedup (since even if the sparsity is fully irregular, the speedup is still significant, see S. Han et al. 2016). Compared with prior accelerators, we have much higher potential for accelerating each VMM (i.e., Y=WX+b) since in DSG both the X from previous layer and the Y in current layer are sparse and the skip of W access is structured (skip whole rows or columns, see Fig. 3(b) and Fig. 4).
>
>   Specifically, the answers to Q1-3 are listed as follows:
>
>   Q1: The primary target of this work is to validate the functionality of DSG, and we only implement it using multi-thread VMMs on CPU at this stage. Detailed reasons can be found in the above explanations. We have clarified this point in the Experiment Setup section of the previous submission.
>
>   Q2: The dense baseline in Fig. 8(b) use multi-thread VMMs on CPU.
>
>
> To be continued in Part II...

---

> > ### Author Response · Authors · 2018-11-16
> > **Responses to Reviewer#1 (Part II)**
> >
> > Q3: This is an interesting question that can we achieve a joint sparsity pattern across a mini-batch. We think it is possible for simple tasks with only FC-layer based architectures, e.g., through sharing pruning index across different samples [1]. However, the index sharing will lead to more accuracy loss, which forms an issue of accuracy-efficiency trade-off. Furthermore, the CONV layer itself is a GEMM even if without batching samples since each sliding window corresponding to a VMM has independent sparsity pattern.
> >
> >   Sharing sparsity index across sliding windows and samples is equivalent to the weight filter pruning because we can consistently skip the access of some weight columns in Fig. 3(b). Whereas, current filter pruning can only work for inference after complicated training with filter selection and expensive retraining with fixed sparsity pattern. To our best knowledge, we did not see this kind of pruning methods being able to optimize the training phase.
> >
> >   We did additional experiments for training MLP on FASHION dataset and VGG8 on CIFAR10 dataset by sharing the same sparsity pattern across samples (also including sliding windows in CNNs) within each mini-batch according to their selection rate (the weight filter being selected less times across sliding windows and samples will be pruned at current training iteration). We found that the MLP accuracy on FASHION with joint sparsity just decreases 1.2% compared to the vanilla DSG under 50% sparsity; while the CNN with joint sparsity pattern will compromise 16.23% accuracy. This evidences our prediction that the joint sparsity can work on simple tasks, and moreover, CNNs are more difficult to utilize this strategy.
> >
> > References:
> > [1] Sun, Xu, Xuancheng Ren, Shuming Ma, and Houfeng Wang. "meProp: Sparsified back propagation for accelerated deep learning with reduced overfitting." arXiv preprint arXiv:1706.06197 (2017).
> >
> >   Q6: The up to 2.3x speedup data can be found in Fig. 7(a) (the last bar). Yes, it is based on VMM implementations and the number of GMACs is for one iteration. Because DSG will not slow down the convergence speed (see the responses to Q5), the operation reduction in each iteration matters for the entire training.
> >
> > (2)	Q4: More Networks on Fig. 8b:
> >   In the revised manuscript, we have added more architectures for Fig. 8(b), including ResNet8 on CIFAR10 and AlexNet on ImageNet. Due to the limited space, we visualized the new results in Fig. 12 (Appendix D). The previous conclusion for Fig. 8(b) still holds for Fig. 12.
> >
> > (3)	Q5: Regarding Convergence:
> >    Usually, sparse training will slow down the overall convergence since the different sparsity pattern at different iterations creates variance. However, we experimentally found that the convergence speed is comparable with the original dense training under our DSG framework. The training curve of both dense and sparse models are shown in Fig. 10(a) and (b) (newly added in Appendix C) for VGG8 on CIFAR10 and ResNet18 on ImageNet, respectively.
> >
> >    To explore the underlying reason, we visualize the inner product distribution before and after dimension reduction through random projection. The newly added Fig. 10(c) presents the pairwise difference between the original high-dimensional inner product and the low-dimensional one. The data are from the CONV5 layer of VGG8. It can be seen that most of the inner product differences are around zero, which implies an accurate approximation capability of the dimension-reduction search. This helps reduce the training variance and avoid training deceleration.
> >
> > (4)	Q7: Compared to Dropout:
> >   In dropout, the important neurons are randomly selected. In contrast, DSG first reduces the data dimension via random projection and then selected important neurons according to the approximated activations in a low-dimensional space with much less computational cost than the original high-dimensional space. The selection of important neurons at each iteration is approximately equivalent to select the neurons with higher activation values, i.e., not fully random.
> >
> >   In the case of low sparsity, both DSG and fully random dropout can achieve good accuracy and even better because of the regularization effect. However, in the case of high sparsity, the fully random selection will significantly compromise the accuracy due to the ablation of too much useful information. In this case, the dimension-reduction search can remain the accuracy much better since it approximately selects neurons according to their importance. Fig. 5(c) in our original submission has shown this phenomenon, wherein the ‘Random’ curve denotes a random dropout.
> >
> > (5)	Fewer Acronyms:
> >   In the revised manuscript, we have reduced the number of acronyms including DRS (dimension-reduction search) and DMS (double-mask selection). Note that we still keep DRS in some figures due to the space limitation but with full name in the figure captions.

---

### Official Review · AnonReviewer3 · 2018-11-02
**Dnn compression and acceleration**

**Rating:** 7
**Confidence:** 2

**Review:**

REVISED: I am fine accepting. The authors did make it a bit easier to read (although it is still very dense). I am also satisfied with related work and comparisons
Summary:
This paper proposes to activate only a small number of neurons during both training and inference time, in order to speed up training and decrease the memory footprint. This works by constructing dynamic sparse graph for each input, which in turn decides which neurons would be used. This happens at each iteration and it does not permanently remove the neurons or weights. To construct this dynamic sparse graph, authors use dimensionality reduction search which estimates the importance of neurons

Clarity:
Overall I found it very hard to follow. Lots of accronyms, the important parts are skipped (the algorithm is in appendix) and it is very dense and a lot of things are covered very shallowly. It would have been better for clarity to describe the algorithm in more details, instead of just one paragraph, and save space by removing other parts.  I would not be able to implement the proposed solution by just reading the paper

Detailed comments.
This reminds me a lot of a some sort of supervised dropout.

My main concern, apart from clarity, is that there is no experimental comparison with any other method. How does it compare with other methods of dnn compression or acceleration?

Also i found the literature review is somewhat lacking. What about methods that induce sparsity via the regularization, or those that use saliency criterion, hessian based approaches like Song Han, Jeff Pool, John Tran, and William Dally. Learning both weights and connections for efficient neural network. NIPS, 2015. , pruning filters Hao Li, Asim Kadav, Igor Durdanovic, Hanan Samet, and Hans Peter Graf. Pruning filters for, efficient convnets. ICLR, 2017.  etc.
Basically i don't understand how it compares to alternative methods at all.

Questions:
How does it run during inference? does inference stay deterministic (there is a random projection step there)

---

> ### Author Response · Authors · 2018-11-16
> **Responses to Reviewer#3**
>
> Thanks for your valuable suggestions and comments. Details can be found as below.
>
> (1)	Presentation Clarity:
> According to your kind suggestions, we revised the manuscript as follows:
> a)	Reduced the number of acronyms including DRS (dimension-reduction search) and DMS (double-mask selection). Note that we still keep DRS in some figures due to the space limitation but with full name in the figure captions.
> b)	Polished the paper to make it clearer and more in-depth, especially moved Algorithm 1 to the main text.
> c)	Adjusted the spacing to make it less dense.
>
> (2)	Comparison with Other Methods:
> As mentioned by Reviewer #1, our work targets at both training and inference,  while most of the previous work focused on the inference. In prior methods, training usually becomes more complicated with various regularization constraints or iterative fine-tuning/retraining. Therefore, it is not very fair to compare with them, which is the reason we did not include the comparison in the original submission.
>
> This time, after considering your advice, we added additional comparisons with several existing compression methods. We just compared the inference pruning, but we should note that our major contribution is on the training side. The results are shown in Table 2 in the newly added section of Appendix D. Because the focus is on comparing with inference compression approaches, we perform DSG based on pre-trained models rather than training from scratch like most experiments in the first submission. Table 2 demonstrates that DSG can achieve a good balance between the operation amount and model accuracy.
>
> (3)	Literature Review:
> Thanks for pointing out the importance of literature review. However, we are sure that the literature you mentioned in the comments have been cited in our first submission (Introduction and Related Work sections). For example, S. Han et al. “Learning both weights and connections for efficient neural network” (NIPS 2015), H. Li et al. “Pruning filters for efficient convnets” (ICLR 2017), regularization based pruning (e.g. W. Wen et al. 2016, Y. He et al. 2017, Z. Liu et al. 2017), saliency criterion based pruning (e.g. S. Han et al. 2015a/b, P. Molchanov et al. 2016, H. Li et al. 2016, Y. Lin et al. 2017a/b, X. Sun et al. 2017, Y. He et al. 2018a), and other optimization based methods (e.g. J. H. Luo et al. 2017, L. Liang et al. 2018) have already been cited in the previous manuscript.
>
> After considering your feedback, we have cited more emerging references (e.g., T. W. Chin et al. 2018, J. Ye et al. 2018, J. H. Luo et al. 2018, Hu et al. 2018, Y. He et al. 2018b) in the revised manuscript.
>
> (4)	Does Inference Stay Deterministic?
> The project matrices are fixed after a random initialization at the beginning of training. Therefore, inference stays deterministic. We have made this point clear in the revised manuscript (Experiment Setup section).

---

### Official Review · AnonReviewer2 · 2018-11-03
**An interesting method to reduce the memory & time cost for both DNN training and inference**

**Rating:** 8
**Confidence:** 3

**Review:**

[Overview]

In this paper, the authors proposed to use dynamic sparse computation graph for reducing the computation memory and time cost in deep neural network (DNN). This method is applicable in both DNN training and inference. Unlike most of previous work that focusing on the reduction of computation during the inference time, this new method propose a dynamic computation graph by pruning the activations on the fly during the training of inference, which is an interesting and novel exploration. In the experiments, the authors performed extensive experiments to demonstrate the effectiveness of the proposed method compared with several baseline methods and original models. It is clear to me that this method helps to reduce the memory cost and computation cost for both DNN training and inference.


[Strengthes]

1. This paper addresses the computational burden in both memory and time from a novel angle than previous network pruning methods. It can be applied to reduce the computation in both network training and inference, but also preserve the representation ability of the network.

2. To endow the network compression in training and inference, the authors proposed to mute the low-activated neurons so that the computations merely happened on those selected neurons.

3. For the selection, the authors proposed a simple but efficient dimension reduction methods, random sparse projection, to project the original activations and weights into a lower-dimensional space and compute the approximated response map in such a lower dimension space, which the selection is based on.

4. The authors performed comprehensive experiments to demonstrate the effectiveness the proposed method for network compression. Those results are insightful and solid.

[Questions]

1. Is the sparsity of each layer the same across the whole network? It would be nice if the authors could perform some ablation studies on varied sparsity in different layers, maybe just with some heuristic methods, e.g., decreasing the sparsity from lower layer to upper layers. As the authors mentioned, higher sparsity causes a larger degradation on deeper network. I am curious that whether there are some better way to set the sparsity.

2. During the training of the network, how the activation evolve? It would be interesting to show how the selected activation changes across the training time for the same training sample.  This might provide some insights on when the activations begin to converge to a stable state, and how it varies layer by layer.

3. Following the above questions, is there any stage that the sparsity can be fixed without further computation for selection. In generally, the training proceeds for a number of epochs. It would be nice if we can observe some convergence on the selected activations and then we can suspend the selection for saving the computation burden.

[Conclusion]

This paper present an interesting and novel approach for network pruning in both training and inference. Unlike most of the previous work, it pruning the activations in each layer though a dimension reduction strategy. From the experiments, this method achieved an obvious improvement for reducing the computation memory and time cost in training and inference stages. I think this paper has prompted a new direction of efficient deep neural network.

---

> ### Author Response · Authors · 2018-11-16
> **Responses to Reviewer#2**
>
> We appreciate you for the positive feedback and your recognition of our contribution on compressing and accelerating both training and inference via dimension reduction search. Your questions are truly insightful which merit further investigation. Our answers are listed as follows:
>
> (1)	Q1: Layer-wise Sparsity Configuration:
> Yes, we set a uniform sparsity across all layers for simplicity in our experiments because training DNNs is costly. After reading your comments, we find it is fascinating to touch more. To explore the sparsity configuration strategy for different layers is possible to produce better accuracy at the same compression level. However, entirely exploring the configuration space is very time consuming and can be enough to start a new story. Considering the limited time budget for rebuttal, we just did a supplementary experiment with a heuristic strategy, i.e., higher sparsity for compute-intensive layers and lower sparsity for the rest layers.
>
> Taking CONV2-CONV6 layers in VGG8 on CIFAR10 as a case study (since other layers occupy fewer operations), the sparsity configuration of 0.5-0.9-0.9-0.9-0.9 achieves 92.09% accuracy, while the configuration of 0.9-0.5-0.9-0.5-0.9 using the mentioned heuristic strategy could reach 92.89%. Both of two configurations have nearly the same operations but different accuracy results, which reflects that sparsity configuration across layers indeed matters.
> In fact, recent works [1, 2] touched this problem by using fast sensitivity test or reinforcement learning, respectively, to automate the layer-wise sparsity configuration. Their results evidence your guess that a smarter pruning strategy probably gives better accuracy. From Fig. 2 in [1] and Fig. 2-3 in [2], we can see that the optimized sparsity configuration presents a non-uniform distribution across layers. The interesting point is that the distribution is not monotonous but presents fluctuation as layer changes. Although it seems quite hard to reveal the underlying reason, we believe your suggestion is a right direction for future work. We have added reference [1] in the revised Related Work section.
>
> (2)	Q2-3: Selection Pattern Evolvement:
> Our previous submission focused more on the method validation of using the dimension-reduction search to select critical neurons for achieving sparse computational graph dynamically. Your comments indeed pose another interesting question: how does the activation selection evolve and can it converge well? We are also curious about the answer.
>
> To explore this question, we did an additional experiment as shown in the Appendix C Fig. 11 in the revised manuscript. We select a mini-batch of training samples as a case study for data recording. Each curve presents the results of one layer (CONV2-CONV6). For each sample at each layer, we recorded the change of binary selection mask between two adjacent training epochs. Here the change is obtained by calculating the L1-norm value of the difference tensor of two mask tensors at two adjacent epochs, i.e., change=batch_avg_L1norm(mask[i+1] – mask[i]). Here the “batch_avg_L1norm” indicates the average L1-norm value across all samples in one mini-batch. As shown in Fig. 11(a), the selection mask for each sample converges as training goes on.
>
> Actually, in our implementation, we inherit the random projection matrix from training and do the same dimension-reduction search in inference. We didn’t try to suspend the selection masks directly. Our concern is that the selection mask varies across samples even if we observed convergence for each sample. As we can see from Fig. 11(b), the different mask tensors between adjacent samples in one mini-batch present significant differences (large L1-norm value) after training. Therefore, it will consume a lot of memory space to save these trained masks for all samples, which is less efficient than conducting on-the-fly search during inference. Index sharing across different samples [3] might be helpful at the cost of more accuracy degradation. We agree that your feedback on the selection convergence is an exciting problem, and we are pleased to study more in the future.
>
> References:
> [1] He, Yihui, Ji Lin, Zhijian Liu, Hanrui Wang, Li-Jia Li, and Song Han. "AMC: AutoML for model compression and acceleration on mobile devices." In Proceedings of the European Conference on Computer Vision (ECCV), pp. 784-800. 2018.
> [2] Xu, Xiaofan, Mi Sun Park, and Cormac Brick. "Hybrid Pruning: Thinner Sparse Networks for Fast Inference on Edge Devices." arXiv preprint arXiv:1811.00482 (2018).
> [3] Sun, Xu, Xuancheng Ren, Shuming Ma, and Houfeng Wang. "meProp: Sparsified back propagation for accelerated deep learning with reduced overfitting." arXiv preprint arXiv:1706.06197 (2017).

---

### Meta-Review · Area_Chair1 · 2018-12-16
**novel approach**

**Confidence:** 5
**Recommendation:** Accept (Poster)

**Metareview:**

This paper proposes a novel approach for network pruning in both training and inference. This paper received a consensus of acceptance. Compared with previous work that focus and model compression on training, this paper saves memory and accelerates both training and inference. It is activation, rather than weight that dominates the training memory. Reviewer1 posed a valid concern about the efficient implementation on GPUs, and authors agreed that practical speedup on GPU is difficult. It'll be great if the authors can give practical insights on how to achieve real speedup in the final draft.